# Adipose-Derived Mesenchymal Stromal Cell Transplantation for Severe Spinal Cord Injury: Functional Improvement Supported by Angiogenesis and Neuroprotection

**DOI:** 10.3390/cells12111470

**Published:** 2023-05-25

**Authors:** Ai Takahashi, Hideaki Nakajima, Arisa Kubota, Shuji Watanabe, Akihiko Matsumine

**Affiliations:** Department of Orthopaedics and Rehabilitation Medicine, University of Fukui, Fukui 910-1193, Japan; aitun@u-fukui.ac.jp (A.T.);

**Keywords:** adipose-derived mesenchymal stromal cell transplantation, acute spinal cord injury, functional recovery, treadmill exercise training, oxidative stress

## Abstract

Mesenchymal stromal cell transplantation alone is insufficient when motor dysfunction is severe; combination therapy with rehabilitation could improve motor function. Here, we aimed to analyze the characteristics of adipose-derived MSCs (AD-MSCs) and determine their effectiveness in severe spinal cord injury (SCI) treatment. A severe SCI model was created and motor function were compared. The rats were divided into AD-MSC-transplanted treadmill exercise-combined (AD-Ex), AD-MSC-transplanted non-exercise (AD-noEx), PBS-injected exercise (PBS-Ex), and no PBS-injected exercise (PBS-noEx) groups. In cultured cell experiments, AD-MSCs were subjected to oxidative stress, and the effects on the extracellular secretion of AD-MSCs were investigated using multiplex flow cytometry. We assessed angiogenesis and macrophage accumulation in the acute phase. Spinal cavity or scar size and axonal preservation were assessed histologically in the subacute phase. Significant motor function improvement was observed in the AD-Ex group. Vascular endothelial growth factor and C-C motif chemokine 2 expression in AD-MSC culture supernatants increased under oxidative stress. Enhanced angiogenesis and decreased macrophage accumulation were observed at 2 weeks post-transplantation, whereas spinal cord cavity or scar size and axonal preservation were observed at 4 weeks. Overall, AD-MSC transplantation combined with treadmill exercise training improved motor function in severe SCI. AD-MSC transplantation promoted angiogenesis and neuroprotection.

## 1. Introduction

Mesenchymal stromal cells (MSCs) are isolated from a variety of tissues, including the bone marrow, adipose tissue, and synovium. It has been indicated that MSC transplantation is useful in the treatment of spinal cord injury (SCI) [1,2]. Previously, we reported an improvement in motor function, pain relief, suppression of spinal cord cavity formation, and neuroprotective effects after the transplantation of bone marrow stromal cells (BM-MSCs) for acute SCI [3,4]. However, we also found that the survival rate of MSCs after transplantation was low and that sufficient therapeutic effects could not be achieved with transplantation alone [5]. Various degrees of hypoxia [6] and oxidative stress [7] occur after SCI; therefore, transplanted cells are exposed to cytotoxic stress and consequently present a low retention rate after transplantation. Adipose-derived MSCs (AD-MSCs) are autologous transplantable materials isolated from subcutaneous adipose tissue [8,9]. AD-MSCs are useful because they can be harvested in large amounts from adipose tissue, contain high levels of stem cells [9,10], and may be resistant to cytotoxic stresses such as oxidative stress [11,12] and hypoxia [13,14]. In our previous study, we found that AD-MSCs are more resistant to hypoxia and oxidative stress than bone marrow stromal cells (BM-MSCs) and have a higher survival rate after transplantation. Furthermore, AD-MSC transplantation causes functional improvement in moderate SCIs, and AD-MSCs exert histological angiogenic and neuroprotective effects [15]. However, the therapeutic effects of AD-MSCs on severe SCI and their specific effects remain unclear. The aims of this study were to assess the (i) functional improvement of severe SCI by AD-MSC transplantation combined with treadmill training; (ii) promotion of angiogenesis and macrophage accumulation, suppression of cavity or scar formation, and axonal protection after treatment; and (iii) increase in cytokine expression in AD-MSCs under oxidative stress using multiplex flow cytometry.

## 2. Materials and Methods

### 2.1. Animals and Ethics Statement

The study protocol was approved by the Life Science Research Laboratory, University of Fukui, Division of Bioresearch, Japan (protocol code R04043). Animal experiments were conducted in accordance with Guidelines for the Proper Conduct of Animal Experiments (published by the Science Council of Japan). We used 80 male Sprague–Dawley (SD) rats, including those whose findings are not reflected. Animal experiments were approved by the Animal Research Committee of the University of Fukui. All rats were handled by the Division of Laboratory Animal Resources. Some parts of this section overlap with reports of previous studies in which similar experiments on rodents with SCI were conducted.

### 2.2. Isolation and Culture of AD-MSCs

AD-MSCs were isolated from the abdominal subcutaneous fat tissue of 8-week-old male SD rats (*n* = 6; Nippon SLC, Shizuoka, Japan). The animals were anesthetized, and their fat tissues were collected and minced using a razor blade. The characteristics of AD-MSCs vary with the harvest site [16,17]; abdominal subcutaneous tissues reportedly have a high concentration of stem cells [18]. The collected adipose tissue was then enzymatically dissociated for 50 min at 37.5 °C (99.5 °F) using 0.075% collagenase type I (Sigma–Aldrich, Saint Louis, MO, USA). The solution was passed through a 70-μm filter and centrifuged at 300× *g* for 5 min. The cells were washed thrice with phosphate-buffered saline (PBS), resuspended, and cultured in Dulbecco’s modified Eagle’s medium (DMEM; Invitrogen, Carlsbad, CA, USA), supplemented with 10% fetal bovine serum (FBS; Invitrogen). The cultures were maintained at 80–90% confluency in a 37.5 °C incubator with 5% CO_2_ and passaged with 0.025% trypsin/ethylenediaminetetraacetic acid (Invitrogen) when required.

### 2.3. Flow Cytometry

The flow cytometric analysis was performed to analyze the expression of specific cell surface markers on AD-MSCs, as described previously [15]. The cells from the third passage were trypsinized into single-cell suspensions and labeled with the following anti-rat antibodies conjugated to fluorochromes: PE-CD11b (0.25 μg/100 μL; BioLegend, San Diego, CA, USA), PE-CD29 (0.25 μg/100 μL; BioLegend), PE-CD34 (5 µL/100 µL; NSJ Bioreagents, San Diego, CA, USA), PE-CD90.1 (10 μL/100 μL; BioLegend), and PE-CD105 (0.25 μg/100 μL; BioLegend). The cells were incubated with each antibody for 45 min on ice. The corresponding mouse isotype antibodies were used as controls (1:5; BioLegend). The cells were analyzed using the fluorescence-activated cell sorting (FACS) instrument (FACSCanto II; BD Biosciences, Franklin Lakes, NJ, USA) according to the manufacturer’s protocol. The percentage of expressed antigen was calculated for 10,000 gated-cell events, and the data were processed (FACSDiva software; BD Biosciences).

### 2.4. Animal Model of SCI

Adult male SD rats, 9–10 week-old, were used in the experiments. They were maintained on a 12-h light/dark cycle and provided with water and food. The rats were anesthetized with an isoflurane (Forane^®^; Abbot Japan, Osaka, Japan) and oxygen mixture (4.0% and 2.5% for induction and maintenance, respectively) and subjected to T9-T10 total laminectomy. A contusion SCI was generated using a commercially available SCI device (Infinite Horizons Impactor; Precision Systems & Instrumentation LLC, Fairfax Station, VA, USA) with an impact force of 250 kD (severe contusion). The rats received manual bladder expression until their recovery from spontaneous voiding.

### 2.5. Animals and Experimental Groups

The rats were divided into five experimental groups.

(i)Severe contusion model treated with AD-MSCs and treadmill exercise training (AD-Ex group; 1 × 10^6^ cells/5 µL PBS, *n* = 16).(ii)Severe contusion model treated with AD-MSCs without exercise training (AD-noEx group; 1 × 10^6^ cells/5 µL PBS, *n* = 22).(iii)Severe contusion model treated with PBS injection and treadmill exercise training (PBS-Ex group; no cells/5 µL PBS, *n* = 16).(iv)PBS-noEx group: no cells/5 µL PBS injection, or treadmill exercise training (*n* = 17).(v)Sham group: only laminectomy without SCI (*n* = 3).

AD-MSC transplantation and PBS injection were performed 3 days after injury.

In our previous study, we found that the appropriate time for MSC transplantation was 3 days after injury [5]. According to this finding, all groups were subjected to AD-MSC transplantation 3 days after SCI in this study. It has been reported that AD-MSCs are resistant to oxidative stress and present better survival even early after transplantation [15]. The injury epicenter of the AD-Ex and AD-noEx group rats, which was clearly observed in the central region of the laminectomy area, was injected with AD-MSCs suspended in 5 µL of PBS. The rats in the PBS-Ex and PBS-noEx groups were injected with 5 µL of PBS without cells. In all groups, the MSC suspension or PBS was injected into the contusion epicenter using a 10-mL Hamilton syringe with a 26-gauge needle attached to an automated micropump. Rats for histological observation (*n* = 24) were transplanted with AD-MSCs fluorescently labeled with PKH26 (Sigma–Aldrich, Saint Louis, MO, USA); labeling with PKH26 was performed per the manufacturer’s protocol.

### 2.6. Treadmill Training

Before the SCI experiment, the rats in the AD-Ex and PBS-Ex groups were habituated for 3 d on a motorized treadmill at 10 m/min for 10 min each day. Three days after the SCI, the rats were trained to walk on a treadmill for 10 weeks, three times a day for 10 min at 2 m/min for 1 week, followed by 4 m/min for the last 9 weeks. The training was performed 5 d/week. In cases where the rat stopped walking or turned around, it was lifted up and manually repositioned on the moving treadmill.

### 2.7. Assessment of the Motor Function

Motor functional outcomes were evaluated using the Basso, Beattie, and Bresnahan (BBB) locomotor scale method, in which the scores ranged from 0 (no hind limb movement) to 21 (complete functional recovery) [19]. The BBB scores of each group were recorded at 7, 14, 21, 28, 35, 42, 49, 56, 63, and 70 d after transplantation by two independent examiners blinded to the experimental conditions (S.W. and A.K.). The groups tested were the AD-Ex, AD-noEx, PBS-Ex, and PBS-noEx groups (*n* = 5 in each-point).

### 2.8. Multiplex Flow Cytometry

AimPlex^®^ (AimPlex Biosclience, Pomona, CA, USA) is a bead-based flow cytometry reagent that enables simultaneous quantification of multiple cytokines from a small sample. Passage 3 AD-MSCs were cultured in six-well plates with 20.5% oxygen at 37.5 °C (99.5 °F) for 6 h at 5000 cells/well. The culture medium was then changed to 0, 250, 500, 750, and 1000 µM hydrogen peroxide-supplemented culture medium (DMEM containing 10% FBS), and the cells were cultured under the same conditions for 24 h. The cells were collected using trypsinization and analyzed for 17 different proteins, following the manufacturer’s protocol.

### 2.9. Immunohistochemistry

Histological evaluation was performed according to our previously described method [15]. Once deep anesthesia was achieved, the AD-Ex, AD-noEx, PBS-Ex, and PBS-noEx group rats (*n* = 6 in each group) were intracardially perfused with PBS, followed by fixation with 4% paraformaldehyde in PBS, and the spinal cords were excised and post-fixed with 4% paraformaldehyde overnight. The samples were then immersed in 10% sucrose in PBS for 24 h and 30% sucrose in PBS for another 24 h. The spinal cord segments (T8 to T12) were embedded using OCT compound (Sakura Finetek, Osaka, Japan), frozen at −80 °C (−112 °F), and cut into 14-µm thick sagittal sections using a cryostat. For immunofluorescence staining with fluorescent antibodies, frozen sections were permeabilized with 0.1 mol/L Tris-HCl buffer (pH 7.6) containing 0.3% Triton X-100. The following primary antibodies were diluted in a commercial diluent (antibody diluent with background-reducing components; Dako Cytomation, Glostrup, Denmark): anti-CD31 monoclonal antibody (1:100, rabbit IgG; Abcam, Waltham, MA, USA, marker for vascular endothelium) and anti-CD11b antibody (1:100, mouse IgG; Sigma–Aldrich, marker for macrophages). The sections were incubated with the primary antibodies overnight at 4 °C. The sections were then incubated for 1 h at room temperature (RT) with Alexa Fluor 488 (1:250; Abcam, Cambridge, UK) as the secondary antibody. Thereafter, the sections were rinsed in PBS three times and examined via epifluorescence. Immunofluorescence images were captured using a confocal laser-scanning microscope (TCS SP2; Leica Microsystems, Wetzlar, Germany). The following procedure was performed to semi-quantify the expression of fluorescent dyes: 3 mid-sagittal sections at the injury epicenter were selected randomly, and 4–6 non-overlapping high-magnification photomicrographs (×200; CD31; ×400; CD11b) were captured per section (TCS SP2; Leica Microsystems). Observations were made at 1 and 2 weeks after transplantation. The fluorescent staining-positive areas were analyzed using NIH ImageJ software.

### 2.10. Assessment of the Spinal Cavity or Scar Size and Axonal Sparing

The cavity or scar formation and axonal protection after SCI were evaluated using hematoxylin and eosin (HE) and Luxol Fast Blue (LFB) staining. The rats in the AD-Ex, AD-noEx, PBS-Ex, and PBS-noEx groups (*n* = 3 in each group) were sacrificed 4 weeks after transplantation. In our previous study, we found that tissue-repair and cavity-suppression effects after MSC transplantation were observed 4 weeks after transplantation [3]. Under anesthesia, the rats were intracardially perfused with PBS, and the samples were fixed with 4% paraformaldehyde in PBS. The tissue samples were then immersed in 20% sucrose for 24 h, and segments of the spinal cord were cut on a cryostat into 14-μm thick axial sections. To quantify the cavity or scar areas and axonal sparing at 4 weeks after transplantation, the following procedures were performed: 10 axial sections, ±500 µm, ±1000 µm, and ±1500 µm from the epicenter, were randomly selected, and high-magnification (×200) photomicrographs of the posterior funiculus were captured (TCS SP2; Leica Microsystems). Positively stained HE and LFB areas were analyzed using ImageJ software.

### 2.11. Statistical Analysis

The results are reported as the mean ± standard deviation (SD). The results of AimPlex^®^ multiplex flow cytometry analysis and histological assessments were examined for statistically significant differences between groups using a one-way analysis of variance (ANOVA). A two-way ANOVA was used to compare the results of the motor function analysis. In the two-way ANOVA, the BBB score was the dependent variable; treatment group (AD-Ex, AD-noEx, PBS-Ex, and PBS-noEx) and time after treatment (1–10 weeks) were independent variables. There were no interactions between independent variables. A *p* value of <0.05 denoted a significant difference in Tukey’s post-hoc analysis. All statistical analyses were performed using SPSS (SPSS Inc., Chicago, IL, USA).

## 3. Results

### 3.1. CD Marker Expression in AD-MSCs

The CD marker expression is shown in Figure 1. The third passage SD rat AD-MSCs isolated from the subcutaneous fat tissue were characterized with MSC surface markers; AD-MSCs were strongly positive for CD29 and CD90.1, but negative for CD11b, CD34, and CD105.

### 3.2. Functional Recovery after AD-MSC Transplantation and Treadmill Exercise Training

In order to confirm that AD-MSCs were securely implanted within the spinal cord, the distribution of the cells after transplantation was observed (Figure 2). AD-MSCs fluorescently labeled with PKH26 diffused in a cranio-caudal direction in the transplanted area from 1 to 2 weeks after transplantation. Motor function was assessed 10 weeks after treatment using the BBB score (*n* = 5 at each time-point). Nine rats died before 10 weeks after SCI (each 2 rats in AD-Ex, AD-noEx, and PBS-Ex, and 3 rats in PBS-noEx). The scores were the highest in the AD-Ex group compared with those in the other groups. Significant differences were observed between these groups at 7 weeks post-transplantation, after which the increase in score decreased. No significant differences were observed among the AD-noEx, PBS-Ex, and PBS-noEx groups, and no change in the score was observed in the sham group with laminectomy only (Figure 3).

### 3.3. Effect of Oxidative Stress on AD-MSCs

The effects of oxidative stress on the extracellular secretion of AD-MSCs were investigated. Seventeen cytokines were measured simultaneously using the AimPlex^®^ multiplex flow cytometry analysis (Figure 4). Both vascular endothelial growth factor (VEGF) and chemokine (C-C motif) ligand 2 (CCL2) were highly expressed, with the strongest expression observed under 250 µM hydrogen peroxide treatment, whereas both proteins were not expressed under 1000 µM hydrogen peroxide treatment.

### 3.4. Angiogenesis and Macrophage Accumulation after AD-MSC Transplantation

The results of fluorescent immunostaining are shown in Figure 5. CD31 and CD11b were used as markers for vascular endothelial cells and macrophages, respectively, and evaluated in the midsagittal section of the spinal cord (Figure 5A,C). At 1 week after transplantation, there were no significant differences among the groups. At 2 weeks after transplantation, more CD31-positive areas and fewer CD11b-positive areas were observed in the AD-MSC transplant group (AD-Ex and AD-noEx groups) than in the non-transplant group (PBS-Ex and PBS-noEx groups) (Figure 5B,D).

### 3.5. Suppression of Cavity or Scar Formation and Axonal Sparing after AD-MSC Transplantation and Treadmill Exercise Training

The results of the histological evaluation are shown in Figure 6. The axial section of the spinal cord was morphologically evaluated using HE staining from the epicenter to the cranio–caudal side at 1500 µm (Figure 6A). Significant suppression of cavity/scar area was observed in the AD-Ex group compared with that in the AD-noEx, PBS-Ex, and PBS-noEx groups. LFB staining of the epicenter showed that the myelin sheath was significantly larger in the AD-Ex and AD-noEx groups than in the PBS-Ex and PBS-noEx groups (Figure 6B).

## 4. Discussion

The benefits of MSC transplantation in the treatment of SCI have been reported [20,21,22,23,24,25,26]. After transplantation, MSCs rarely differentiate, have a very low risk of tumorigenesis, and are thought to exert their therapeutic effects through unique exocrine function [27,28]. However, MSCs disappear within a few weeks when transplanted directly into the injury site [3]. Bone marrow-derived MSCs (BM-MSCs) have mainly been used as a source of MSCs for the treatment of SCI. However, in recent years, AD-MSCs have attracted attention as an alternative autologous transplant material, mainly because of their relative ease of isolation [9,29] and their high stem cell content [10]. There are detailed reports on the treatment of SCI using AD-MSC transplantation [30,31]. We previously reported that AD-MSC transplantation has functional-improvement effects on moderate SCI [15]; however, the effect of AD-MSCs on severe SCI was not clarified. The reasons for this were the difficulty involved in a detailed assessment of motor function in the mouse model and the short observation period. In the present study, the rats were treated in four groups and observed for lower limb movements. Motor function was assessed over a 7-week period. Other studies have shown that AD-MSCs exhibit angiogenic [32] and neuronal regenerative effects [33]. In this study, we evaluated the following: (i) whether transplantation of AD-MSCs and treadmill training resulted in functional improvement in severe SCI; (ii) whether oxidative stress enhanced AD-MSC cytokine secretion; and (iii) histological effects after AD-MSC transplantation: angiogenesis, macrophage accumulation pattern, cavity or scar suppression, and axonal repair. The results of this study suggested the following regarding the therapeutic effects of AD-MSCs on SCI: (i) the combination of AD-MSC transplantation and treadmill training significantly improved motor function in severe SCI; (ii) the exocrine effects of AD-MSCs are enhanced under oxidative stress; (iii) AD-MSC transplantation promotes angiogenesis and inhibits macrophage accumulation in the acute phase; (iv) AD-MSC transplantation inhibits scar and cavity formation and has neuroprotective effects in the subacute phase.

Before the experiments, we confirmed the expression of the surface markers on third-passage AD-MSCs from SD rats using flow cytometry. The cells used in this study were positive for CD29 and CD90 but negative for CD11b, CD34, and CD105, consistent with the findings of a previous study [34]. Anderson et al. reported that a subpopulation of MSCs is negative for CD105, but its expression increases with passage [35]. Among the surface antigens investigated in this study, CD34, one of the hematopoietic markers, has interesting properties: although it is a surface marker of AD-MSCs in humans [36,37], its concentration decreases over time [38] and is differentially expressed among rodent species [39].

In the present study, direct transplantation was chosen to ensure transplantation of cells into the injury site; AD-MSCs labeled with PKH26 were observed in the spinal cord after transplantation and distributed cranio caudally during the observation period. This phenomenon supported the findings of our study using bone marrow-derived MSCs [3]. As a behavioral assessment, a severe SCI model corresponding to the American Spinal Injury Association (ASIA) impairment scale [40] (an international classification of SCI; A–E, A corresponds to complete motor-sensory dysfunction) grades A–B was created using an IH impactor. Our model showed complete motor paralysis lasting several days after injury. Patients with severe SCI classified as ASIA grades A–C usually present with permanent motor-sensory impairment, which may interfere with daily activities. Ten-week behavioral observations showed that AD-MSC transplantation combined with treadmill training resulted in significant improvements in motor function compared to AD-MSC transplantation without training and PBS injection with/without training. The results of this experiment showed that AD-MSC transplantation alone did not improve motor function in severe SCI and that post-transplantation rehabilitation led to an improvement in the BBB score. Our previous study results showed an improvement in motor function 4 weeks after transplantation for moderately severe SCI [15]. However, a longer observation period was set in this study because of the severe injury, and motor function improvement was observed 7 weeks after treatment. The exercise training of the rats was carried out with care to avoid individual differences and ethical issues. Although there are several rehabilitation programs for rats [41,42,43], pre-training was performed before the injury, and two levels of exercise load were applied after transplantation to ensure that there were no differences in adherence to training and reduce stress by applying a load appropriate to the degree of recovery.

Next, we conducted a bead-based immunoassay that allowed the simultaneous quantification of multiple cytokines from a small sample volume. We quantified the changes in the expression of 17 proteins upon the oxidation of AD-MSCs. The results showed that the expression of VEGF and CCL2 was upregulated under 250 µM hydrogen peroxide treatment and gradually decreased up to 1000 µM hydrogen peroxide treatment. VEGF specifically affects vascular endothelial cells and induces angiogenesis and lymphangiogenesis [44,45]. Furthermore, it has neurotrophic and neuroprotective effects on glial cells and neurons in the central and peripheral nervous systems [46]. The type of VEGF detected in this study was not identified because of the nature of the reagents used. CCL2 is a migration factor for monocytes and basophils; it has tissue-repair and anti-inflammatory effects [47]. Sierra-Filardi et al. reported that CCL2 has anti-inflammatory effects through the promotion of interleukin-10 production [48]. The present study results suggest that AD-MSCs are characterized by a humoral factor that increases under oxidative stress.

Histological evaluations were performed, focusing on changes over time after SCI. Local inflammation, ischemia, and oxidative stress are the main causes of secondary injury after direct external force. Angiogenesis and macrophage accumulation were assessed using immunofluorescence staining at 1–2 weeks, the acute phase of SCI. AD-MSC-transplanted areas showed a strong expression of CD31 in the vascular endothelium, supporting our previous findings. The first peak of macrophage accumulation after SCI has been reported to occur between 3 and 7 days after injury [49]. In the present results, a higher number of macrophages were observed in the first week after injury, supporting previous study findings. Furthermore, there were both beneficial and detrimental effects on macrophage accumulation after SCI [50], suggesting that AD-MSC transplantation modulates macrophage accumulation.

Spinal cavity or scar formation was assessed using HE staining and axonal sparing using LFB staining at 4 weeks, which is the subacute phase at which inflammation subsides and tissue repair occurs. Inflammatory scarring and cavity formation inhibit axonal regeneration [51]. The AD-MSC-transplanted groups showed suppressed cavity/scar formation and promoted axonal sparing. Interestingly, the functional recovery is not entirely consistent with these histological changes; although histological repair was observed in the AD-MSC-transplant group with/without exercise, transplantation alone did not improve motor function. Recovery of motor function required both AD-MSC transplantation and treadmill training. The discrepancy between image findings and physical function is often experienced in real-world situations in rehabilitation medicine.

## 5. Conclusions

AD-MSC transplantation combined with treadmill exercise training improved motor function after severe SCI. In in vitro experiments, the expression of VEGF and CCL2 in AD-MSCs was increased by oxidative stress. Histological assessment showed that transplantation of AD-MSCs in the acute phase promoted angiogenesis and inhibited macrophage accumulation; in the subacute phase, they inhibited scar/cavity formation and promoted tissue repair. These findings suggest that AD-MSCs are an effective transplant material for severe SCI.

## Figures and Tables

**Figure 1 cells-12-01470-f001:**
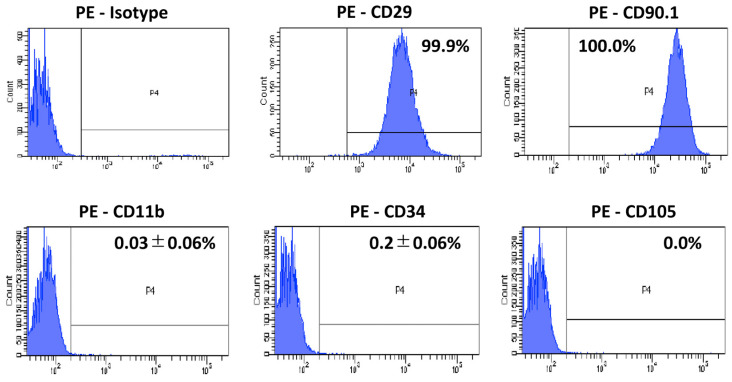
CD marker expression in AD-MSCs. The third-passage SD rat AD-MSCs were positive for CD29 and CD90.1, but negative for CD11b, CD34, and CD105.

**Figure 2 cells-12-01470-f002:**
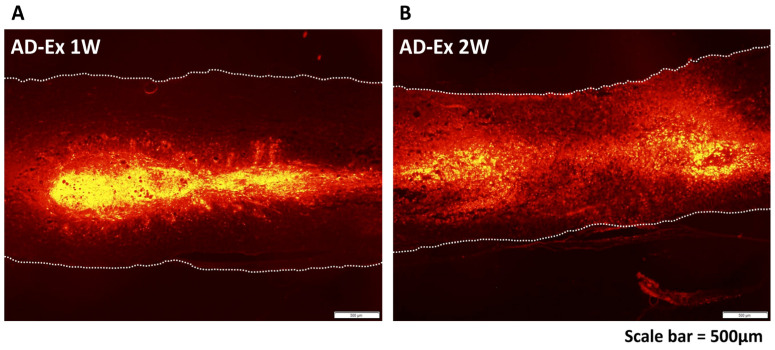
AD-MSC distribution after transplantation. Distribution of AD-MSCs labeled with PKH26 at 1 week (**A**) and 2 weeks (**B**) after transplantation. The transplanted cells were spread in a cranio-caudal direction within the spinal cord.

**Figure 3 cells-12-01470-f003:**
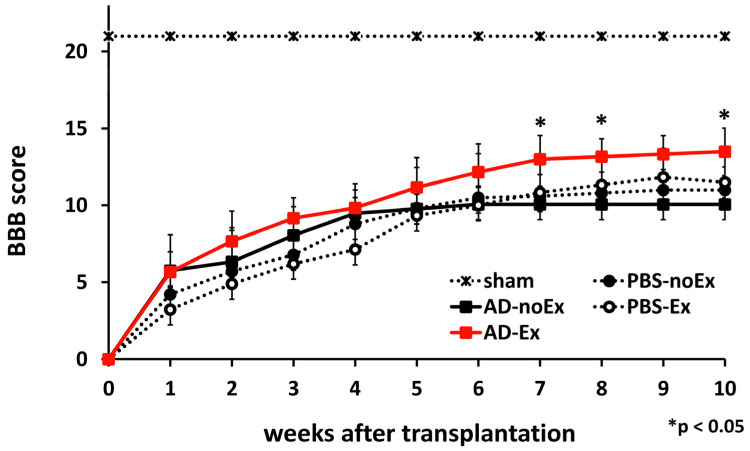
Motor function assessment. The BBB score of the AD-MSC transplantation with treadmill training (AD-Ex) group indicated significantly better motor function than that of the other groups; there were no significant differences between the AD-MSC transplantation without training (AD-noEx) and PBS injected with/without training (PBS-Ex/PBS-no Ex) groups. Data are expressed as mean ± SD. * *p* < 0.05. AD-MSC: adipose-derived mesenchymal stromal cell; BBB: Basso, Beattie, and Bresnahan Scale; SD: standard deviation.

**Figure 4 cells-12-01470-f004:**
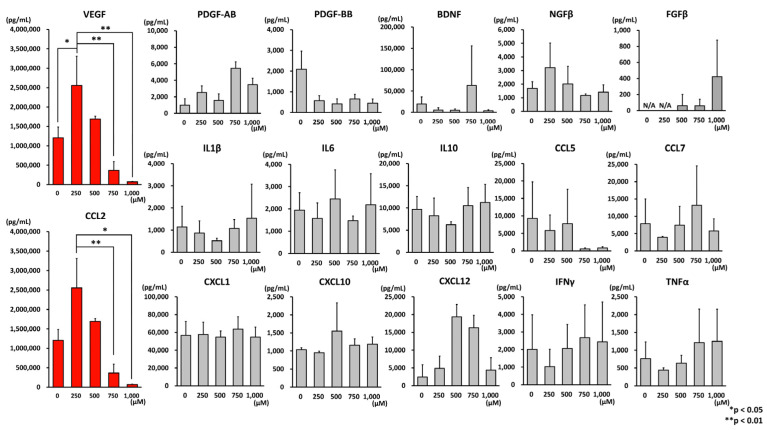
Results of multiplex flow cytometry analysis. AD-MSCs were cultured under hydrogen peroxide-added conditions and analyzed using multiplex flow cytometry for 17 cytokines/chemokines in the culture supernatant. Both VEGF and CCL2 were highly expressed in the culture supernatant of AD-MSCs. The strongest expression of these cytokines was observed under 250 µM hydrogen peroxide treatment, but under 1000 µM hydrogen peroxide treatment, both proteins were not expressed. Data are expressed as mean ± SD. * *p* < 0.05; ** *p* < 0.001. VEGF: vascular endothelial growth factor; CCL2: C-C motif chemokine 2; SD: standard deviation.

**Figure 5 cells-12-01470-f005:**
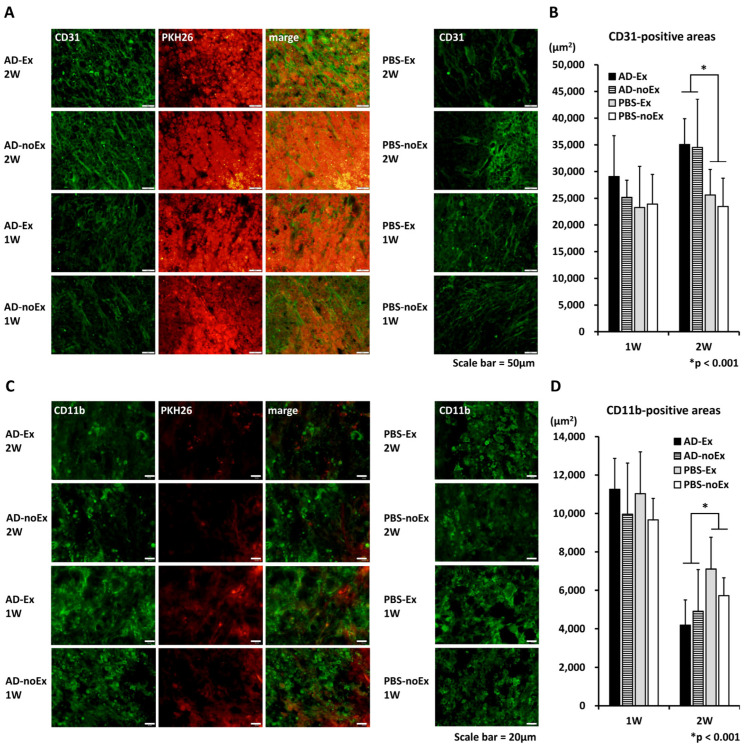
Results of immunofluorescence staining. There were no significant differences among the groups at 1 week after transplantation. Two weeks after transplantation, more CD31-positive (**A**,**B**), and fewer CD11b-positive areas (**C**,**D**) were observed in the AD-MSC transplant group (AD-Ex and AD-noEx groups) than in the non-transplant group (PBS-Ex and PBS-noEx groups). Data in the graph are expressed as mean ± SD. * *p* < 0.001. SD: standard deviation.

**Figure 6 cells-12-01470-f006:**
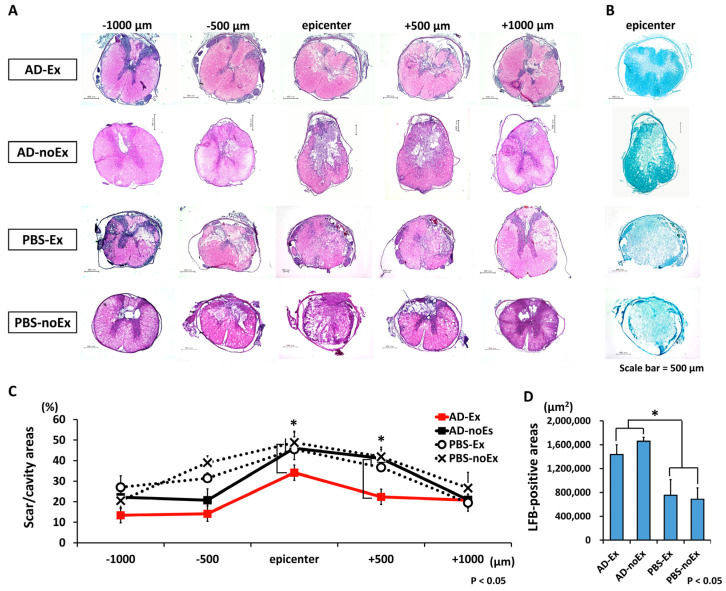
Results of histological assessment. HE staining of the axial section of the spinal cord of each group (**A**) and LFB staining of the epicenter (**B**). HE staining at 4 weeks after transplantation showed significantly less cavity and scar formation in the AD-Ex group than in the other groups (AD-noEx, PBS-Ex, and PBS-noEx) (**C**). LFB staining of the epicenter showed that the myelin sheath area was larger in the AD-MSC transplant groups (AD-EX and AD-noEx) than in the PBS-injected groups (PBS-Ex and PBS-noEx) (**D**). Data in the graph are expressed as mean ± SD. * *p* < 0.05. HE: hematoxylin and eosin; LFB: Luxol Fast Blue; SD: standard deviation.

## Data Availability

Raw data were generated at University of Fukui. Derived data supporting the findings of this study are available from the corresponding author A.T. and H.N. on request.

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
