# Peer review of "Adipose-Derived Mesenchymal Stromal Cell Transplantation for Severe Spinal Cord Injury: Functional Improvement Supported by Angiogenesis and Neuroprotection"

_cells, 2023, doi:10.3390/cells12111470_

Round 1
Reviewer 1 Report
Summary
Mesenchymal stromal cells (MSCs) face high oxidative stress conditions, and often die after transplanted in the injured spinal cord. Combination therapies can be used to limit the negative effects observed in intraspinal MSC transplantation alone. This study examined the efficacy of adipose-derived MSCs (AD-MSCs) combined with treadmill exercise in a rat model of severe contusive SCI. Rats were divided into the following groups: AD-MSC-transplanted treadmill exercise-combined (AD-Ex), AD-MSC-transplanted non-exercise (AD-noEx), PBS-injected exercise (PBS-Ex), and no PBS-injected exercise (PBS-noEx). Acute phase post-injury tissue was immunofluorescence labeled to evaluate macrophage aggregation and extent of vasculogenesis. In the subacute phase after injury, lesion cavity, scar size, and axonal protection were assessed histologically. Longitudinal functional assessments of treatment efficacy were also performed. Functional ability was significantly increased by AD-MSC-Ex treatment. Angiogenesis in the injured cord may be explained by the increased expression of VEGF and C-C motif chemokines 2 in AD-MSC secretions, as the value of these factors increased in under oxidative stress. A significant reduction in these pathologies was observed in the combined cell/exercise treatment group at four weeks post-SCI. The authors concluded from their study’s findings that AD-MSC transplantation combined with treadmill exercise exhibited neurological and histological neuroprotective benefits, and that oxidative stress induced VEGF and chemokine production may explain these therapeutic effects. This article is fairly well written and advances our understanding of combination therapies to overcome stem cell transplant limitations in SCI treatment. However, there are some significant issues that need to be addressed and these are listed below.
Major
1. In the Materials and Methods section on Animals and Ethics Statement (line 64) it is not stated whet sex of rat was used for the study. This is an important piece of information and should be included in this section and the section 2.4 Animal Model of SCI. Also, the protocol number and institute guidelines should be stated, and that these were followed in the completion of the study.
2. In the Materials and Methods section on Isolation and Culture of AD-MSCs, it should be clearly stated what region of the body the subcutaneous fat was collected from. Different regions of fat yield stem cells of different characteristics and paracrine secretions. Documenting the specific area and why this area was selected for isolation will help future researchers better interpret and replicate the study.
3. In the Abstract, it says that the Ad-MSC secretions in the supernatant were analyzed by flow cytometry, but the Materials and Methods section on Flow Cytometry stated the cell were labeled for the markers and analyzed by FC (which makes more sense). The authors should make these areas consistent. Also, in this section in the Materials and Methods, the authors should clearly describe why they analyzed these specific markers. Were they used to ensure these were true Ad-MSCs?
4. What were the Ad-MSCs transplanted into the injured cored 3 days post injury? If the authors have a rationale for this, they should state it. Usually, cells are transplanted at 7 days post-injury or later as this is when the major inflammatory and secondary injury cascades have downregulated and there is space in the cord to increase survival of the cells and limit damage to the cord from transplantation.
5. Were the cells injected by hand or stereotaxically?
6. It is a bit confusing how the subgroups of animals were assigned. In the Abstract and results/figures it says histological analysis was performed up to 4 weeks post SCI, but the functional analyses continued out to 70 days. All that is stated in the Materials and Methods is that “Rats for histological observation (n = 24) were transplanted with AD-MSCs fluorescently labeled with 120 PKH26 (Sigma-Aldrich, MO, USA); the method of labeling with PKH26 followed the manufacturer’s protocol.” However, adding up the animals from Sec. 2.5 only comes to 50 rats, not 98 as stated in Sec. 2.1. The authors need to address this and better described how the animal groups and subgroups were divided to account for the total number of animals, and this should match all statements of animal group numbers in the paper.
7. For the Statistical Analysis section the authors state that both a Student’s t-test and one-way ANOVA were used for analyzing data. First, I do not see any data that indicates a Student’s t-test would be used as there are multiple groups, and t-test only measures a single timepoint set of data between two groups. Secondly, especially for the longitudinal functional data, the correct test for this would be a repeated measures two-way ANOVA. You have two factors, time and treatment group, and you analyze the same animals' function at multiple times. I suggest these data be reanalyzed using the correct statistical tests and the correct tests, then describe these correctly in the Materials and Methods.
Minor
8. There are two periods following 2.5 in line 102. This is also observed following 2.9 in line 146 and 2.10 (line 171) of the Materials and Methods.
9. In the beginning section of the Discussion, thought the authors do described the relevant prior studies they are comparing their findings too, the number of studies and references appears limited, in light of the many MSC transplantation studies that have been performed and published in SCI models.
Author Response
The point-by-point response is in Word file format.

Reviewer 2 Report
The work you show is good and without any problem for its publication, I liked the way of presenting the results that are congruent with the suggested hypothesis.
Perhaps the only thing I would do as an extra is in the evaluation of the area of injury or scar in addition to the stains that you used to check the decrease in damage, I would use and quantify GFAP.
Author Response
The work you show is good and without any problem for its publication, I liked the way of presenting the results that are congruent with the suggested hypothesis.
Perhaps the only thing I would do as an extra is in the evaluation of the area of injury or scar in addition to the stains that you used to check the decrease in damage, I would use and quantify GFAP.
We thank you for this important suggestion. GFAP is appropriate for neuroregenerative and glial scar assessments. However, GFAP may also stain new nerve tissues before myelination. In this study, we used LFB staining to assess myelinated area. Your suggestion will be considered in our future study.
Round 2
Reviewer 1 Report
The authors have addressed my concerns and questions concerning this manuscript.